# Cancer patients' needs assessment in primary care: study protocol for a cluster randomised controlled trial (cRCT), economic evaluation and normalisation process theory evaluation of the needs assessment tool cancer (CANAssess)

Joseph Clark [1], Bethan Copsey [2], Alexandra Wright-Hughes [3], Emma McNaught,[3] Petra Bijsterveld,[3] Terry McCormack,[4] Robbie Foy [5], Scott Wilkes,[6] Jon Mark Dickson [7], David Meads,[5] Amanda Farrin,[3] Miriam Johnson [1]

For numbered affiliations see end of article.

**Correspondence to**
Dr Joseph Clark;
joseph.clark@hyms.ac.uk

## ABSTRACT

**Introduction** Unmet needs in patients with cancer and their carers are common but poorly identified and addressed. The Needs Assessment Tool-Cancer (NAT-C) is a structured consultation guide to identify and triage patient and carer unmet needs. The NAT-C is validated, but its effectiveness in reducing unmet patient and carer needs in primary care is unknown.

**Methods and analysis** Cluster randomised controlled trial with internal pilot and embedded process evaluation to test the clinical and cost effectiveness of the NAT-C in primary care for people with active cancer in reducing unmet patient and carer need, compared with usual care. We will recruit 1080 patients with active cancer (and carers if relevant) from 54 general practices in England. Participating practices will be randomised 1:1 to either deliver an NAT-guided clinical consultation plus usual care or to usual care alone. Consenting participants with active cancer and their carers (if nominated) will be asked to complete study questionnaires at baseline, 1 and 3 months for all, 6 months except for those recruited outside of the last 3 months of recruitment, and attend an NAT-C appointment if allocated to an intervention practice. An internal pilot will assess: site and participant recruitment, intervention uptake and follow-up rates. The primary outcome, the proportion of patients with an unmet need on the Supportive Care Needs Survey Short Form 34 at 3 months postregistration, will be analysed using a multilevel logistic regression. Mixed-methods process evaluation informed by Normalisation Process Theory will use quantitative survey and interview data from clinicians and key stakeholders in cancer care to develop an implementation strategy for nationwide rollout of the NAT-C if the intervention is cost-effective.

**Ethics and dissemination** Ethical approval from London-Surrey REC (20/LO/0312). Results will be peer-reviewed, published and made available to research participants.

**Trial registration number** ISRCTN15497400.

## Strengths and limitations of this study

⇒ We are testing the clinical and cost-effectiveness of the Needs Assessment Tool-Cancer, which has been validated and shown to be acceptable to patients and clinicians during feasibility testing.

⇒ Feasibility testing led to modifications of intervention delivery and informed trial design, making successful completion of the trial more likely.

⇒ CANASSESS is a cluster randomised controlled trial of 54 general practices across 2 regions in England, making it likely that findings will be generalisable nationwide.

⇒ By necessity, participants, health professionals delivering the intervention and study researchers will be aware of treatment allocation; potential bias will be monitored during the trial.

⇒ COVID-19 presents unique challenges in terms of safely conducting clinical trials in primary care.

## INTRODUCTION

Unmet needs in people with cancer and their carers are common but poorly identified and addressed. Many people with cancer experience unmet needs across multiple domains.[1] General Practitioners (GPs, family doctors) and other clinicians in primary care would like to do more to support their cancer patients, but there is no agreed evidence-based best approach.[2] Difficulties are compounded by inconsistent co-ordination of care with oncology services as GPs may be unaware of problems unless patients present directly. However, people with cancer often do not attend primary care for cancer care and systematic, routine holistic assessment of

patient problems is rare.[3] In addition, patients commonly volunteer only the most pressing problem to their clinicians; open enquiry in one study only found an average of one problem presented, whereas systematic enquiry discovered an average of ten, many of which were severe and distressing.[4]

Tools are available to assist clinicians caring for people with cancer,[5] but few are designed to identify and triage care needs in the everyday busy clinical setting and across all stages of active disease from diagnosis through to end of life care. Furthermore, although needs assessment tools are advocated,[6] there is no rigorous research evidence to indicate whether they actually improve practice and patient outcomes. A needs assessment tool can reduce unmet needs by providing a consistent and comprehensive approach to prompting discussion patients' range of support and care needs; helps professionals triage tailored action and is useful for audit and service planning.[7–11] Through triage, an assessment tool may help reduce late referrals for palliative care, and improve referrals where there are physical, psychological, social and spiritual problems.[12 13] However, tools currently available are commonly highly detailed and long for daily clinical use.[14–16]

### Development of the Needs Assessment Tool Cancer

The Needs Assessment Tool-Cancer (NAT-C) was developed in Australia, where it has been shown to reduce unmet needs of patients in oncology clinics.[3] We adapted and validated this tool for use in UK primary care.[17] Use of the NAT-C aims to reduce unmet supportive and palliative care needs of cancer patients and their carers by supporting systematic clinician assessment of patient and carer needs across multiple domains. Identified problems may be managed in primary care or through referral to other services.

Our phase II feasibility study found that a randomised trial is feasible in terms of recruitment, data quality and intervention delivery.[18] Required changes to improve study processes were identified, specifically, confirmation of participant acceptability to be directed to a known NAT-C clinician. Our Resource Use Questionnaire (RUQ) was also modified following feedback from patient participants in the feasibility study. Clinicians, patients and carers also viewed the tool positively and supported need for a definitive trial. A key alteration to the NAT-C was to develop the paper-based tool into digital templates for use in standard electronic clinical record systems (EMIS, SystmOne) in accordance with clinician preferences.

### Aims

The CANAssess trial aims to evaluate the clinical effectiveness and cost effectiveness of the NAT-C in reducing unmet needs of patients and carers in primary care carer compared with usual care alone.

## METHODS AND ANALYSIS
### Design summary

CANAssess is a multicentre, two-arm, pragmatic, cluster randomised controlled trial (cRCT) with 12-month internal pilot, embedded process evaluation and cost-effectiveness evaluation. A cRCT design reflects that the intervention would be implemented at general practice level and reduces contamination in the control group.

The trial opened to recruitment on 1 October 2020, recruitment is expected to cease on 1 June 2022 and participant follow-up will end on 1 September 2022.

Trial objectives and outcomes are reported in box 1.

### Recruitment setting

The study aims to recruit patients and their carers from 54 general practices (clusters) from four geographical regions (recruitment 'hubs') in Yorkshire, East Midlands and the North East of England. Locations were selected to ensure a range of multi-ethnic, rural and urban populations to maximise generalisability of findings.

### Recruitment of general practices

Site identification and recruitment is detailed in figure 1. General practices will be eligible unless they: took part in the feasibility study, have or are planning to implement within the duration of the trial a systematic holistic cancer care intervention that overlaps with the NAT-C, or lack capacity and capability to deliver the study.

### Cluster randomisation

Where practice manager agreement is obtained, capacity and capability confirmed, and initial read-code search completed, participating general practices (clusters, n=54) will be randomised sequentially via an automated system at the clinical trials research unit (CTRU). General practice randomisation will be 1:1 to: implement the NAT-C in addition to usual care, or usual care alone, using a computer-generated minimisation programme incorporating a random element to ensure arms are balanced for stratification factors:
▶ Locality: Urban or rural area.[19 20]
▶ List size: <5000, 5000–10000, >10000.[20]
▶ A GP training practice (obtained from site feasibility questionnaire): yes, no.

General practices and research nurses providing participant recruitment and follow-up support across multiple surgeries will, by necessity, be aware of treatment allocation. However, no member of the research team will be involved with intervention delivery to minimise performance bias. A structured risk of bias assessment is presented in online supplemental file 1. Participating practices will be free to withdraw from the study without negative consequence. In the event of practice withdrawal, we will inquire about reasons for withdrawal and may recruit replacement practices.

### Participant eligibility

Eligibility criteria are shown in box 2

## Box 1 CANAssess Primary, Secondary, internal pilot, economic and process evaluation objectives

### Primary objective
To test the effectiveness of the Needs Assessment Tool-Cancer (NAT-C) compared with usual care in reducing unmet patient need as measured using the Supportive Care Needs Survey Short Form 34 (SCNS-SF34)[33] at 3 months postregistration.

### Secondary objectives
To evaluate the effectiveness of the NAT-C compared with usual care with regard to:
⇒ Patient unmet need on psychological, health system information, physical and daily activity, patient care and support, and sexuality domains of the SCNS-SF34 at 1, 3 and 6 months.
⇒ Patient performance status, measured using the Australian-modified Karnofsky Performance Status[34] at 1, 3 and 6 months.
⇒ Patient severity of symptoms, measured using the Revised Edmonton Symptom Assessment System[35] at 1, 3 and 6 months.
⇒ Patient mood and quality of life as measured by the European Organisation for Research and Treatment of Cancer Quality of Life-C15-Palliative questionnaire[14] at 1, 3 and 6 months.
⇒ Carers' ability to care and carer well-being as measured using the Carer Experience Scale[36] and Zarit Burden Interview-12[37] at 1, 3 and 6 months.
To evaluate intervention delivery, uptake and fidelity of the NAT-C as measured by:
⇒ NAT-C training of general practitioners and nurses in each general practice.
⇒ Completed NAT-C consultations by patient and general practice (including completion of individual items of the NAT-C).
⇒ Length of NAT-C consultations.
⇒ Referral patterns and actions taken to meet identified unmet need (including referrals to health professionals and/or services) from the completed NAT-C.

### Internal pilot objectives
To assess sufficiency of numbers of general practices and patients at 12 months post start of recruitment, we will proceed with the trial unchanged if we have 80% (43) sites open and are recruiting to 80% (48 participants per month) of target. We will assess intervention uptake, follow-up rates and potential for selection bias.

### Health economic objectives
Service utilisation, referral patterns and cost-effectiveness measured using:
⇒ Bespoke Resource Use Questionnaire for capturing patient healthcare service utilisation and referral patterns at 1, 3 and 6 months.
⇒ The EQ-5D-5L,[38] ICEpop CAPability Supportive Care Measure[39] and CES to generate quality-adjusted life-years and estimates of well-being at 1, 3 and 6 months.

### Process evaluation objectives
To assess the adequacy of NAT-C training, intervention fidelity, possible mechanisms of action and issues regarding implementation in practice if the intervention is effective.

## Participant recruitment
General practices will identify eligible patients by searching cancer registers and screening for eligibility. Eligible patients will be sent a letter with a patient information sheet and expression of interest form. General practices may also send an SMS text message or amended letter to patients inviting them to express interest in the study on the CANAssess website. Patients will provide informed consent (online supplemental file 2) ahead of registration into the study. Consented patients may nominate carers for participation in the trial. Carers agreeing to participate will provide consent. The full process of participant recruitment is presented in figure 2. For any participant or carer who wishes to withdraw from the trial, we will collect a reason for withdrawal and cease data collection, but keep collected data unless otherwise requested.

### Intervention arm (NAT-C plus usual care)
The NAT-C comprises five sections: priority referral for further assessment, patient well-being, ability of carer or family to care for patient, carer/family well-being and resulting referrals (if required). Clinicians will be encouraged to use the tool as an aide memoire, conducting a holistic patient assessment as usual, but referring to the NAT-C to ensure all domains are addressed during a consultation. The NAT-C will be completed using either the electronic medical record template (EMIS, SystmOne) or on paper. Completed paper copies of the NAT-C will be uploaded to the patient record.

At least two clinicians per practice will be trained to use the NAT-C either face to face, via webinar or online using a training package piloted during feasibility work.

Participating patients at intervention arm surgeries will be offered a 20 min appointment or home visit depending on clinical need, guided by an NAT-C trained clinician using the tool within approximately 2 weeks of study registration. Appointments will take place either at the practice, at patients' homes or remotely via phone or video according to clinical judgement and coronavirus guidelines. Participating carers will be welcome to accompany patients to their appointment, however, the NAT-C allows assessment of carer need through patient response.

### Usual care
Usual care is defined as management normally provided for patients with cancer registered at the general practice concerned.[21]

### Data collection
Required data, assessment tools, collection time points and processes are summarised in table 1.

### Baseline assessments
Clinical data including comorbidities, cancer stage and treatments will be collected at baseline by the research nurse from the participant's medical record. Demographic information will be collected on participants, including age, sex, participant ethnicity and living arrangements, during the researcher baseline discussion. For carers, age, sex, relationship status and living arrangements will be collected.

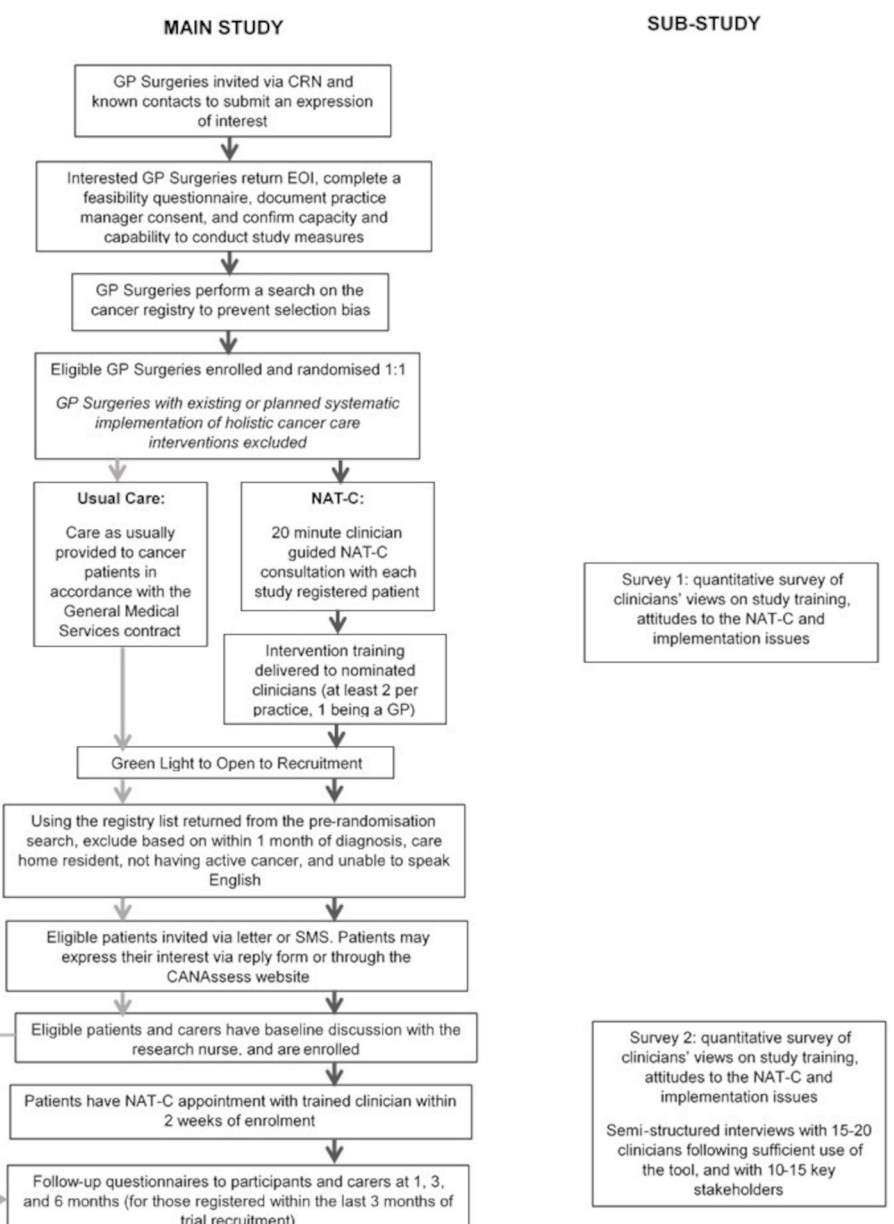

MAIN STUDY

SUB-STUDY

**Figure 1** Study flow chart. CRN, Clinical Research Network; GP, general practitioner; NAT-C, Needs Assessment Tool-Cancer. Expression of Interest (EOI), Short Message Service (SMS)

### Participant questionnaires

Self-reported participant and carer outcome measures will be collected via questionnaires at baseline, 1-month and 3-month postregistration. Questionnaires will also be collected at 6 months for participants and carers registered before 3 months prior to the end of participant recruitment.

Participants will be able to complete questionnaires using paper forms sent by post, online via Research Electronic Data Capture (REDCap) or with a researcher over the phone or face-to-face, as appropriate. Only CTRU data and statistical staff will have direct access to the dataset.

Researchers will telephone participants to confirm questionnaire receipt and assess and collect performance (AKPS) and COVID-19 status.

### Intervention data collection

A research nurse will collect information on NAT-C intervention delivery and content, including the timing, duration, mode of delivery, referrals and subsequent appointments from the participant's medical record.

### Safety data collection

In this population, it is expected that episodes of acute illness, infection, new medical problems and deterioration of existing medical problems will occur and could result in prolonged hospitalisation, hospital readmission, significant or permanent disability or incapacity, or death.

Only serious adverse events fulfilling the definition of a related unexpected serious adverse event resulting from administration of any research procedure, and participant

## Box 2  Patient/carer inclusion/exclusion

**Patient inclusion criteria**
⇒ Adults (aged 18 years and above).
⇒ Diagnosis of active cancer (receiving anticancer treatment both with curative or palliative intent; managed with 'watch and wait'; recurrent or metastatic; or inoperable).
⇒ Willing and able to complete questionnaires at the trial follow-up schedule.
⇒ Provision of written or observed verbal informed consent.
⇒ Sufficient knowledge of the English language to provide informed consent and complete trial questionnaires. The use of an appropriate translator/interpreter is allowed.

**Patient exclusion criteria**
⇒ Patients in complete remission (no clinical or radiological evidence of cancer, and at least 1-month post anticancer treatments).
⇒ Patients with basal cell carcinoma.
⇒ Patients living in a care home or other institutional setting.
⇒ Patients within 1 month of receiving their initial cancer diagnosis.

**Carer inclusion criteria**
⇒ Adults (aged 18 and above).
⇒ Nominated by participant.
⇒ Able to complete trial measures.
⇒ Written or observed verbal informed consent.
⇒ Sufficient knowledge of the English language to provide informed consent and complete trial questionnaires. The use of an appropriate translator/interpreter is allowed.

**Carer exclusion criteria**
⇒ Employed to look after the participant.

deaths during the trial period, will be recorded. Survival status of participants will be ascertained by research nurses from general practices ahead of sending study follow-up questionnaires.

### Deaths
The date and cause of all deaths occurring during the trial period (to last participants 3-month follow-up assessment) will be collected by the researcher from participant's medical record.

### STATISTICAL CONSIDERATIONS
#### Sample size
The study has been powered to detect improvement in patients' level of unmet need as measured by proportion of patients reporting at least one moderate or high need in domains of the Supportive Care Needs Survey Short Form 34 (SCNS-SF34).[22]

Assuming that the proportion of patients with an unmet need on any SCNS-SF34 domain will be similar to that observed pre-intervention by Waller 2012[3]: 64%, then a sample size of 1080 patients recruited from approximately 54 general practices (540 patients, 27 practices per arm), will provide 85% power with a 5% significance level to detect a relative difference of 22% in the proportion of patients with an unmet need. This is an absolute difference of 14%, from 64% to 50%.

The sample size assumes: a 20% loss to follow-up rate by 3 months, to account for eligible patients who are, or are nearing, end of life; an intracluster correlation coefficient (ICC) of 0.05; an average general practice size of 20; and an adjustment to account for variable practice sizes of 4–40. Given heterogeneity in the design of palliative care services and availability of resources through general practices, and median ICCs reported for outcome variables (0.03) and primary care settings (0.045), an ICC of 0.05 will be used.[23]

### Internal pilot and progression criteria
The internal pilot will end 12 months from recruitment of the first general practice. Data from participants in the internal pilot will be included in the main study analysis.

Progression criteria for recruitment are shown in table 2, based on a traffic-light system of green (go), amber (review) and red (stop), and has been agreed by an independent trial steering committee (TSC) and funder. The TSC will be provided with descriptive data, presented by arm and by general practice to assess internal pilot progression criteria, adherence to the intervention and follow-up, and selection bias at approximately 12 months after the start of the recruitment to inform a decision on continuation of the trial. The internal pilot will not lead to any changes to data collection or the intervention and data from participants in the internal pilot will be included in the main study analysis.

### Statistical analysis
There are no planned interim analyses; outcome data will be analysed once only. All analyses will be conducted on the intention-to-treat (ITT) population, in which all general practices and participants will be included in the analysis according to the group which the GP practice was randomised, and regardless of non-adherence to the intervention or withdrawal from the study. A two-sided 5% significance level will be used for statistical endpoint comparisons.

The flow of patients and general practices through the trial will be presented in a Consolidated Standards of Reporting Trials diagram.

As appropriate for cluster trials recruiting participants after randomisation,[24] statistical testing of baseline participant data will be at the end of the internal pilot and at the end of the study to assess for selection bias.

Analyses of primary (overall unmet need) and secondary outcomes (unmet needs, severity of symptoms, quality of life, carer well-being and burden) will use multilevel logistic or linear regression (as appropriate) with participants nested within general practices, and general practices treated as a random effect. The model will be adjusted for the following fixed effects: GP practice-level stratification factors, important participant-level covariates (eg, baseline unmet need, age, sex, cancer status, baseline performance status), and other relevant known predictors of outcome. Results will be expressed as point estimates, p values, ICCs and 95% CIs.

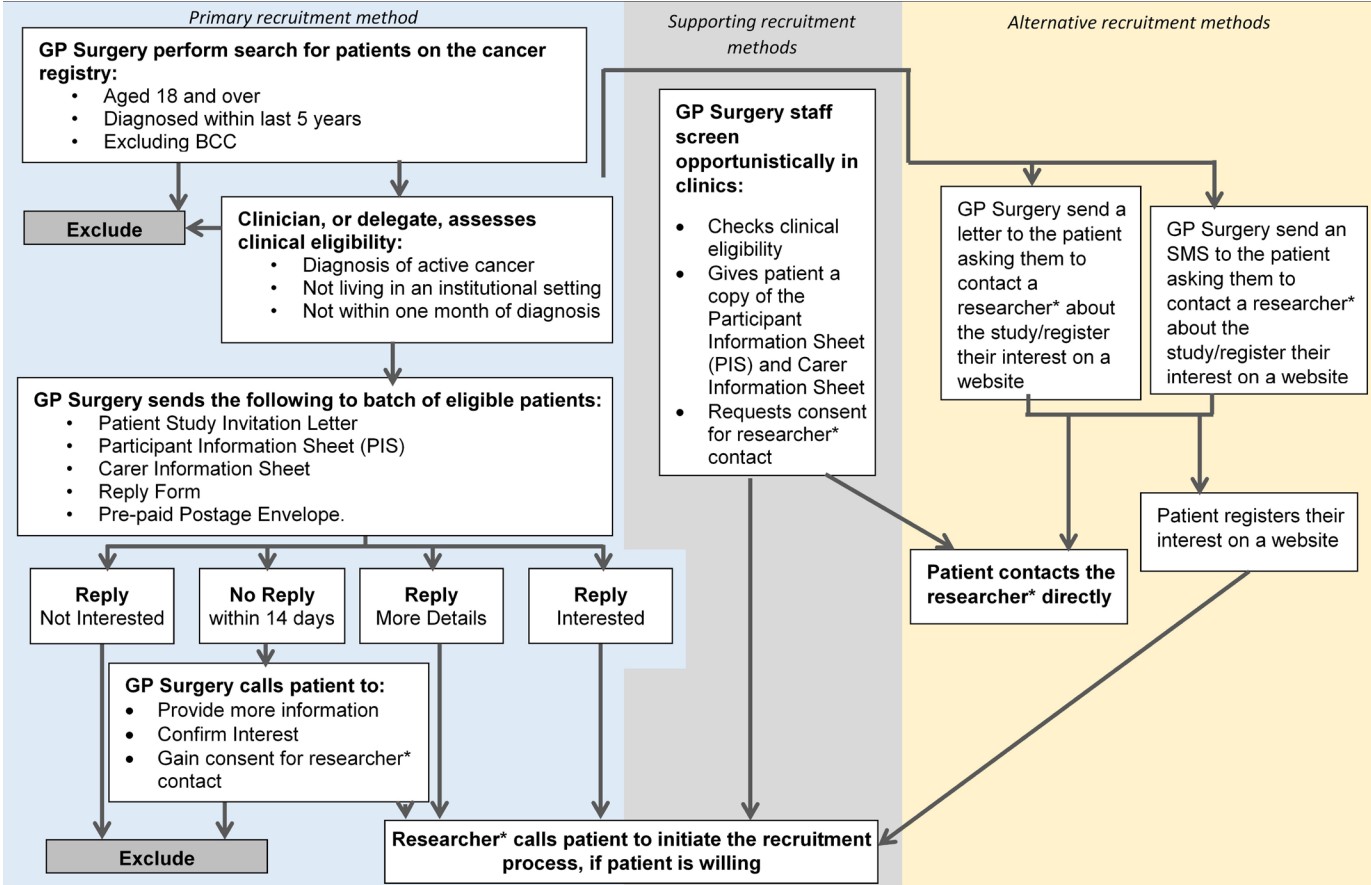

**Figure 2** Participant and carer recruitment. GP, general practitioner; BCC, Basal Cell Carcinoma.

Reasons for attrition and missing participant data will be summarised and mechanisms for missing data we explored according to participant characteristics, intervention and control groups.[25] To conduct analysis on the ITT population, missing data will be multiply imputed at individual participant level under the missing at random assumption. Sensitivity analyses of the primary endpoint will be conducted to assess impact of missing data, choice of imputation model and missing at random assumption.

Quantitative summaries for AKPS score and corresponding change from baseline will be presented at baseline and months 1, 3 and 6 by treatment group. Intervention delivery will be summarised overall and by general practice to evaluate uptake of the NAT-C, adherence to the processes and quality of intervention delivery.

## ECONOMIC EVALUATION

Within-trial health economic evaluation will be undertaken to assess cost-effectiveness of NAT-C vs usual care. The cost–utility analysis will be conducted alongside the trial and follow National Institute for Health and Clinical Excellence reference case for health technology appraisals.[26] The main health outcome will be quality-adjusted life-years (QALYs) based on the Euroqol- 5 Dimension (EQ-5D-5L) (base case). Supplementary analyses will estimate cost

per improvement in ICEpop CAPability Supportive Care Measure (ICECAP-SCM) and Carer Experience Scale (CES).

We will fully cost intervention delivery and measure service utilisation using a bespoke RUQ and measure outcomes using the EQ-5D-5L, ICECAP-SCM and CES at 1, 3 and 6 months.

A patient-completed RUQ will gather data on community-based (eg, contact with GPs, nurses and physiotherapists/occupational therapists), specialist palliative care (hospice, hospital or community) and hospital-based (eg, A&E visits and hospital attendances) healthcare resource utilisation at follow-up. Participants will be given a diary planner to keep to note any healthcare attendances to facilitate completion of the RUQ. Costs will be estimated using UK NHS reference unit costs, data from the Personal Social Services Research Unit and British National Formulary. The primary perspective is the health and personal social service provider but a secondary analysis will adopt a wider perspective to incorporate costs and productivity loss incurred by patients and carers.

Results will be presented as incremental cost-effectiveness ratio (ICERs). Results will also be presented as expected net monetary benefit and cost-effectiveness acceptability curves based on non-parametric bootstrapping.[27] The analysis will employ regression models to adjust for baseline imbalances and account for the correlation between costs and QALYs.[28]

**Table 1** Summary of assessments*

| Participant assessment (including who is involved) | Timeline (months postrandomisation) | | | |
| --- | --- | --- | --- | --- |
| | Baseline | 1 | 3 | 6 |
| Eligibility and consent | | | | |
| Consent (P, C, R) | X | | | |
| Eligibility (assessed by clinician, R) | X | | | |
| Background and demographics | | | | |
| General demographics (P, C, R) | X | | | |
| Cancer demographics (R-case notes) | x | | | |
| Comorbidities (R-case notes) | X | | | |
| Follow-up data (collected from case notes) | | | | |
| Survival status (R) | Ongoing and at the overall end of the trial | | | |
| Related unexpected serious adverse events (R) | Ongoing | | | |
| NAT-C Intervention (R) | One month post participant registration | | | |
| Usual care data (R) | X | X | X | X |
| Prequestionnaire (phone call at 1, 3, 6 months) | | | | |
| Performance status (AKPS) | x | x | x | x |
| COVID status | x | x | x | x |
| Participant Questionnaire Booklet | | | | |
| (Self-completion with researcher support if needed) | | | | |
| Unmet needs (SCNS-SF34) | x | x | x | x |
| Symptoms (ESAS-r) | x | x | x | x |
| Mood and Quality of Life (EORTC QLQ-C15-PAL) | x | x | x | x |
| EuroQol-5 Dimension L (EQ-5D-5L) | x | x | x | x |
| ICECAP-SCM | x | x | x | x |
| Healthcare Resource Use (including usual care data and referrals) | x | x | x | x |
| Carer Questionnaire Booklet | | | | |
| (Self-Completion with researcher support if needed) | | | | |
| Carer Experience Scale | x | x | x | x |
| Carer well-being and burden (ZBI-12) | x | x | x | x |

*P, participant; C, carer-giver; R, researcher.
AKPS, Australian-modified Karnofsky Performance Status; EORTC QLQ-C15-PAL, European Organisation for Research and Treatment of Cancer Quality of Life-C15-Palliative; EQ-5D-L, EuroQol-5 Dimension L ; ICECAP-SCM, ICEpop CAPability Supportive Care Measure; NAT-C, Needs Assessment Tool-Cancer; SCNS-SF34, Supportive Care Needs Survey Short Form 34; ZBI-12, Zarit Burden Interview-12.

The analysis will assume a willingness to pay threshold of £20 000 per incremental QALY with ICERs below this value indicating cost effectiveness.

## PROCESS EVALUATION

A mixed-methods substudy will use normalisation process theory (NPT) to structure data collection and analysis of: (1) implementation of the NAT-C in trial general practices and (2) clinicians' and staff perspectives on the usefulness and effectiveness of the NAT-C, how this relates to usual care and how, if effective, the NAT-C could be implemented nationwide.

NPT is a well-established framework for understanding the dynamics involved in implementing, embedding and integrating a new intervention. We will draw on quantitative and qualitative elements to identify issues related

**Table 2** Progression criteria for internal pilot

| Criteria | Green (go) | Amber (review) | Red (stop) |
| --- | --- | --- | --- |
| Recruitment General practices assessed at 12 months | 80% open (≥43) | 50%–80% open (27-42) | <50% open (<27) |
| Recruitment Participants per month assessed at 12 months (target after 3 months: 60 per month) | ≥80% (≥48) | 50%–80% (30-47) | <50% (<30) |

to implementation in terms of (1) a quantitative NPT survey to elicit the views of clinicians who have undergone NAT-C training and (2) qualitative interviews/focus groups with general practice staff, clinicians and external stakeholders with key roles in health policy and commissioning, relevant to cancer care in primary care.

### Normalisation MeAsure Development Questionnaire survey

The NPT survey (NoMAD instrument) is a 23-item instrument for measuring implementation processes from the perspective of professionals directly involved in the work of implementing complex interventions. During feasibility testing, we adapted the NoMAD instrument in to a 17-point checklist to specifically address the NAT-C. Clinicians will be invited to complete the NoMAD survey either on paper or online following completion of NAT-C training (survey 1). Using results from survey 1, emerging qualitative findings and experiences, the NoMAD will be adapted to include questions regarding emerging issues and concerns. At the end of a practices' involvement with the study, clinicians who have used the NAT-C will be asked to complete the adapted NoMAD survey (survey 2).

Clinicians will be asked questions on a Likert scale in relation to issues such as: attitudes to the NAT-C, NAT-C training and implementation concerns. Completion of the survey will imply informed consent. Data collection and management for surveys 1 and 2 will be delivered by the University of Hull (UoH). All survey data will be anonymised.

### Interviews and focus groups

Opinion regarding NAT-C training, the role and place of the NAT-C within routine practice will be sought from clinicians who received NAT-C training and experts from a range of stakeholder groups (eg, local commissioning groups, general practice federations, the National Cancer Research Institute's primary care group, Royal College of GPs, and Macmillan). Semistructured interviews and focus groups using a priori topic guides (either phone/video conferencing or face to face, as appropriate) will be conducted at various time points post-NAT-C use and up to the end of study. Interviews/focus groups with clinicians and key stakeholders will focus on structural and policy issues relevant to potential implementation of the NAT-C in general practices nationwide, should trial results be positive.

Maximum variation purposive sampling will be used to optimise exploration of a range of clinicians, practice staff and key stakeholder perspectives. An initial purposive sampling grid for clinicians (profession, years of clinical practice, randomisation strata) will be expanded with further criteria identified from implementation study survey responses.

A sample of 15–20 clinicians and general practice staff and 10–15 experts from a range of stakeholders will be sought through interviews or focus group.

Potential interviewees will be provided with a study invitation, a study information sheet and asked to provide informed written consent prior to study procedures. All interviews and focus group discussions will be audiorecorded.

### NoMAD survey analysis

Free-text responses in survey 1 will be monitored by the implementation study researcher to enable rapid feedback to inform subsequent training at other sites.[29]

Once all surveys 1 and 2 are completed, free-text responses will be subject to thematic analysis and descriptive statistics will be used to analyse Likert scale responses including: (1) the extent to which the intervention fits with current practice in relation to the components of NPT; (2) the potential relevance of the NAT-C to individuals' roles; (3) adequacy of NAT-C training and (4) clinician attitudes to the NAT-C at baseline and at the end of the trial from survey 1 and 2.

### Interview/focus group analysis

Qualitative data will be analysed using thematic analysis,[29] informed by NPT, relating to: how clinicians understand the intervention (coherence); how they engage with it (cognitive participation); enact it (collective action) and appraise its effects (reflexive monitoring).[30] The end of trial analysis will develop themes in relation to how the NAT-C could be implemented in primary care nationally, should trial be results be positive. Transcripts will be coded line by line.

### Synthesis with intervention uptake data

We will synthesise key aspects of process evaluation data, with effectiveness of the NAT-C within clusters according to randomisation strata, to improve understanding using NPT about how and if the NAT-C should be implemented into clinical practice using critical interpretative synthesis.[31]

Kirkpatrick's model for training evaluation will be used to evaluate NAT-C training in terms of: reaction to the training, learning and skills improvement, behavioural change and results.[32] Reaction will be assessed by responses to NoMAD surveys and interview. Learning and behavioural change will be evaluated through qualitative data.

### Trial organisation and governance

CANAssess is sponsored by the UoH coordinated by Leeds CTRU and UoH. The sponsor had no direct input in to the design or conduct of the study. The Trial Management Group consists (TMG) of coapplicants, trial coordinators, four GP-hub leads and a public–patient representative. The TMG is responsible for clinical setup, ongoing management, promotion of the trial, and for the interpretation and publishing of the results. A TSC will meet annually and on request to provide independent oversight of the trial and reports to the Sponsor.

A Data Monitoring and Ethics Committee is not needed due to the nature of the study. The TSC will adopt a safety monitoring role, with the constitution of a subcommittee to review safety issues where necessary.

## Patient and public involvement

An experienced lay representative was part of our funding application. She also reviewed and edited public-facing study documentation, and sits on our TMG, with public–patient involvement as a standing item. A further lay representative forms part of our TSC.

## ETHICS AND DISSEMINATION
### Dissemination

If trial results are positive, the NAT-C has the potential to become the gold standard cancer care delivery in primary care as the only valid tool subjected to formal effectiveness testing.

Findings will be presented and discussed at a final dissemination meeting, to which a wide range of stakeholders will be invited, including trial clinicians, participants and those involved in the stakeholder engagement.

Results of the study will be published in peer-review publications and will be presented at national and international conferences. A lay summary of our findings will be published on study and organisational websites, sent to participating general practices and will be accessible to participants.

## Ethical considerations

The trial received ethical approval from the London-Surrey REC (20/LO/0312). Any future amendments to the trial will be submitted to the REC and participants will be informed of any changes which may affect them.

## Impact of COVID-19

The COVID-19 outbreak in England occurred just as ethical approval for the study had been obtained and the process of site identification had begun. We halted site identification and adapted the trial processes to allow remote intervention delivery as per practice procedure for remote consultations, telephone consent and data collection, and online patient study responses and online completion of follow-up questionnaires. Amidst concerns that patient recruitment may be affected by social distancing measures, the Leeds CTRU also highlighted how their secure online computer systems would allow online informed consent provision and data collection. We; therefore, submitted an amendment to allow all study activity to be completed remotely through phone or videoconference.

## Trial status

Following COVID-19-related delays, the trial team is in place, incorporating employed trial-specific research nurses and Clinical Research Network support. Recruitment of GP practices and participants is underway. Our first study site was opened for recruitment on 21 October 2020 and we now have seven general practices recruiting participants. The first participant was recruited on 1 December 2020. As of 25 January 2022, we have 27 general practices open to recruitment and have recruited 333 patient participants and 102 carer participants. This manuscript has been prepared in accordance with study protocol V.3, 24 June 2020. A copy of the full protocol is available on request from JC.

**Author affiliations**
[1]Wolfson Palliative Care Research Centre, University of Hull, Hull, UK
[2]Leeds Institute of Clinical Trials Research, University of Leeds Clinical Trials Research Unit, Leeds, UK
[3]Clinical Trials Research Unit, University of Leeds, Leeds, UK
[4]Hull York Medical School, University of Hull, Hull, UK
[5]Leeds Institute of Health Sciences, University of Leeds, Leeds, UK
[6]School of Pharmacy, University of Sunderland, Sunderland, UK
[7]Academic Unit of Primary Medical Care, The University of Sheffield, Sheffield, UK

**Contributors** All authors meet the International Committee of Medical Journal Editors (ICMJE) criteria for authorship and contributed to the work presented as follows: conception and design of the trial (JC, AW-H, DM, RF, SW, JMD, TM, AF, PB, EM and MJ), development of data analysis methods (AF, BC, DM and AW-H), process evaluation methods (JC/MJ). JC produced a first draft the manuscript, after which all authors commented and provided edits ahead of finalisation. All authors approved the final draft and agree to be held accountable for all aspects of the work by ensuring that questions related to the accuracy and integrity of the work are appropriately investigated and resolved.

**Funding** This work was supported by Yorkshire Cancer Research, grant number: H423.

**Competing interests** None declared.

**Patient and public involvement** Patients and/or the public were involved in the design, or conduct, or reporting, or dissemination plans of this research. Refer to the Methods section for further details.

**Patient consent for publication** Not applicable.

**Provenance and peer review** Not commissioned; externally peer reviewed.

**ORCID iDs**
Joseph Clark http://orcid.org/0000-0003-1410-0996
Bethan Copsey http://orcid.org/0000-0001-9783-6549
Alexandra Wright-Hughes http://orcid.org/0000-0001-8839-6756
Robbie Foy http://orcid.org/0000-0003-0605-7713
Jon Mark Dickson http://orcid.org/0000-0002-1361-2714
Miriam Johnson http://orcid.org/0000-0001-6204-9158

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
