## [Reviewer comments · BMJ Open]

ARTICLE DETAILS

TITLE (PROVISIONAL)	CANcer Patients' Needs Assessment in Primary Care: Study Protocol for a Cluster Randomised Controlled Trial (cRCT), economic evaluation and Normalisation Process Theory evaluation of the Needs Assessment Tool Cancer (CANAssess)
AUTHORS	Clark, Joseph; Copsey, Bethan; Wright-Hughes, Alexandra; McNaught, Emma; Bijsterveld, Petra; McCormack, Terry; Foy, Robbie; Wilkes, Scott; Dickson, Jon; Meads, David; Farrin, Amanda; Johnson, Miriam

VERSION 1 – REVIEW

REVIEWER	Campbell, Karen Edinburgh Napier University, School of health and Social Care
REVIEW RETURNED	20-Aug-2021

GENERAL COMMENTS	Thank you for allowing me to review your protocol. This is a comprehensive protocol which is underpinned by the authors' prior published feasibility study. Initially my concern was focused on the plethora of practice evaluation studies being conducted in the UK around implementation of Holistic Needs Assessment in acute and primary care settings. This is addressed in part by excluding those practices that have engaged in improvement projects implementing HNA's. I would also ask the research team to consider acute sector activity on implementing eHNA as an influencing factor. The research tools appear appropriate for both the economic analysis and the normalization theory and RCT. It will be important to report the feasibility of the scaling of the research project and the relevance of the needs assessment in securing palliative care referrals.
--

REVIEWER	Merriel, Samuel University of Exeter, College of Medicine & Health
REVIEW RETURNED	25-Aug-2021

GENERAL COMMENTS	Thank you for asking me to review this protocol paper of the CANAssess study. The manuscript is well structured, clearly written, and presents the key relevant information regarding the trial. This trial will extensively assess the NAT-C tool from a clinical, patient, economic, and implementation perspective, and could potentially change practice to improve the care of patients with cancer in primary care. There are three issues the authors may want to consider and address: 1. As highlighted in the Introduction, patients with active cancer and those undergoing treatment often receive some of their care
---

	from their GP surgery and some from their oncology service. This intervention is very much primary care focused, and it is unclear whether/how it will be assessed if care delivered by a patient's oncology service impacts on the effectiveness of the intervention and/or the implementation. This may well be covered in the process evaluation, but it was not clear in the manuscript if that is the case. 2. The proposed time to deliver the intervention by GPs is 20 minutes. I recognise the pilot study showed this was the median time taken in intervention practices, but as a practicing GP this seems a relatively short amount of time to allow for what could be an extensive consultation. Have allowances been made for longer consultations both in the surgery or home visits where needed? 3. Regarding the internal pilot, will there be any changes to the intervention or trial design for the full trial as a result of data gathered in the internal pilot? If so, combining internal pilot with full trial data may not be comparing the same intervention. If not, suggest making this clear to the reader to avoid concerns about the analysis plan.
--	---

REVIEWER	Carrera, Pricivel DKFZ, Digital Cancer Prevention
REVIEW RETURNED	20-Dec-2021

GENERAL COMMENTS	It was a pleasure reviewing the manuscript entitled “CANcer Patients’ Needs Assessment in Primary Care: Study Protocol for a Cluster Randomised Controlled Trial (cRCT), economic evaluation and Normalisation Process Theory evaluation of the Needs Assessment Tool Cancer (NAT-C)” for the BMJ Open. The protocol for CANAssess was well-written and study well-designed but needs to address a few points before it could be accepted for publication. First, given the authors’ evaluation of existing tools and their implementation, it is rather surprising that a recent systematic review of 18 RCTs and 8 NRCTs on the effectiveness and safety of screening of psychosocial well-being and care needs of people with cancer was not mentioned in that the findings could inform the design of the study (Schouten B, et al. Cochrane Database Syst Rev. 2019 Mar 26;3(3):CD012387. doi: 10.1002/14651858.CD012387.) Related to this, and since the authors talk about the NAT-C being (potentially) the gold standard, it would be interesting to note which relevant tools (i.e. NCCN Distress Thermometer (DT), the DT-Parent, the Psychosocial Assessment Tool 2.0, the Pediatric Quality- of- Life Inventory, the Children’s Depression Inventory, or the Patient- Reported Outcomes Measurement Information System (PROMIS)) actually implement as part of routine care. Third, for purposes of transparency, the authors need to specify the four (4) geographical regions (recruitment “hubs”) in Yorkshire and the North East of England that they will recruit patients and their carers and how they will ensure “a range of multi-ethnic, rural and urban populations to maximise generalisability of findings” (lines 11-14, pp.6). Fourth, further elaboration of how insights from the feasibility study informed the CANAssess especially in collecting the data for the cost-effectiveness analysis is needed unless its contribution to the final design of the protocol/study is minimal. A minor point, meanwhile, is to clarify in the protocol whether recruitment has started (https://ctru.leeds.ac.uk/wp-content/uploads/2021/08/CANAssess_Participant_Follow-up-Newsletter_v2.0_20200420_August.pdf). A last point concerns the
--

	conduct of a debriefing with participants including the GPs that will be taking part in the study.
--	--

VERSION 1 – AUTHOR RESPONSE

Reviewer 1, Dr. Karen Campbell, Edinburgh Napier University	Thank you for allowing me to review your protocol. This is a comprehensive protocol which is underpinned by the authors' prior published feasibility study.	Thank you	
	5	Initially my concern was focused on the plethora of practice evaluation studies being conducted in the UK around implementation of Holistic Needs Assessment in acute and primary care settings. This is addressed in part by excluding those practices that have engaged in improvement projects implementing HNA's. I would also ask the research team to consider acute sector activity on implementing eHNA as an influencing factor.	See response to point 9
	6	The research tools appear appropriate for both the economic analysis and the normalization theory and RCT.	Thank you
	7	It will be important to report the feasibility of the scaling of the research project and the relevance of the needs assessment in securing palliative care referrals.	Thank you. We are collecting referral information from the NAT-C form, including to palliative care services. In Box 1 (p5) we state with reference to evaluation of intervention of delivery we will measure: "Referral patterns and actions taken to meet identified unmet need (including referrals to health professionals and/or

			services) from the completed NAT-C.”
8	Reviewer 2, Dr. Samuel Merriel, University of Exeter	Thank you for asking me to review this protocol paper of the CANAssess study. The manuscript is well structured, clearly written, and presents the key relevant information regarding the trial. This trial will extensively assess the NAT-C tool from a clinical, patient, economic, and implementation perspective, and could potentially change practice to improve the care of patients with cancer in primary care.	Thank you
9		There are three issues the authors may want to consider and address: 1. As highlighted in the Introduction, patients with active cancer and those undergoing treatment often receive some of their care from their GP surgery and some from their oncology service. This intervention is very much primary care focused, and it is unclear whether/how it will be assessed if care delivered by a patient's oncology service impacts on the effectiveness of the intervention and/or the implementation. This may well be covered in the process evaluation, but it was not clear in the manuscript if that is the case	Quality of secondary care is out of scope of our trial. Randomisation should distribute experiences of secondary care evenly between the two arms of the study. Our prospective trial does not explore any relationship between quality of secondary care and the intervention.
10		2. The proposed time to deliver the intervention by GPs is 20 minutes. I recognise the pilot study showed this was the median time taken in intervention practices, but as a practicing GP this seems a relatively short amount of time to allow for what could be an extensive consultation. Have allowances been made for longer consultations both in the surgery or home visits where needed?	Clinicians receiving intervention training commonly share concerns regarding appointment length. During study training, we emphasise feasibility of delivery demonstrated by our feasibility study and clinicians are encouraged to use their judgement as to whether to allow a study appointment to overrun where necessary. Participating clinicians are assured that we record actual appointment length and use this to inform any recommendations for implementation through the process evaluation. Our process evaluation also explores views study training (see p12).

			Also whilst not of relevance to this protocol paper – during our development of the tool, its validation and our feasibility trial, the median duration of a NAT consultation was 15 – 20 minutes (Clark et al, A cluster randomised trial of a Needs Assessment Tool for adult Cancer patients and their carers (NAT-C) in primary care: A feasibility study. Plos One , 2021. . https://doi.org/10.1371/journal.pone.0245647).
11		3. Regarding the internal pilot, will there be any changes to the intervention or trial design for the full trial as a result of data gathered in the internal pilot? If so, combining internal pilot with full trial data may not be comparing the same intervention. If not, suggest making this clear to the reader to avoid concerns about the analysis plan.	Our internal pilot aims to assess adequacy of general practice and study recruitment. Acceptability of the intervention was explored as part of our feasibility work and no changes will be made to the intervention during the trial. We will report any changes to our protocol related to recruitment processes as part of our CONSORT statement, when trial results are written up. Data from participants in the internal pilot will be included in the main study analysis. For clarity, We have added the following text: “The internal pilot will not lead to any changes to data collection or the intervention, and data from participants in the internal pilot will be included in the main study analysis.”
12	Reviewer: 3, Dr. Pricivel Carrera, DKFZ, Healtimpact: Health/Economic Insights-Impact	It was a pleasure reviewing the manuscript entitled “CANcer Patients’ Needs Assessment in Primary Care: Study Protocol for a Cluster Randomised Controlled Trial (cRCT), economic evaluation and Normalisation Process Theory evaluation of the Needs Assessment Tool Cancer (NAT-C)” for the BMJ Open. The protocol for CANAssess was well-written and study well-designed but needs to address a	Thank you

		few points before it could be accepted for publication.	
13		First, given the authors' evaluation of existing tools and their implementation, it is rather surprising that a recent systematic review of 18 RCTs and 8 NRCTs on the effectiveness and safety of screening of psychosocial well-being and care needs of people with cancer was not mentioned in that the findings could inform the design of the study (Schouten B, et al. Cochrane Database Syst Rev. 2019 Mar 26;3(3):CD012387. doi: 10.1002/14651858.CD012387.)	Thank you. This study was published after we received funding and our outcome measurement tools had been agreed following our feasibility work. We note the findings of the review and this will be a helpful study with which to interpret the results of our trial.
14		Related to this, and since the authors talk about the NAT-C being (potentially) the gold standard, it would be interesting to note which relevant tools (i.e. NCCN Distress Thermometer (DT), the DT- Parent, the Psychosocial Assessment Tool 2.0, the Pediatric Quality- of- Life Inventory, the Children's Depression Inventory, or the Patient-Reported Outcomes Measurement Information System (PROMIS)) actually implement as part of routine care.	Thank you. The NAT-C is not an outcome measure, but a clinical consultation guide (and therefore not an equivalent to the tools mentioned). If the NAT-C is shown to be effective, it will be a subsequent discussion as to how the implementation of the NAT-C can be evaluated for clinical effectiveness in practice.
15		Third, for purposes of transparency, the authors need to specify the four (4) geographical regions (recruitment "hubs") in Yorkshire and the North East of England that they will recruit patients and their carers and how they will ensure "a range of multi-ethnic, rural and urban populations to maximise generalisability of findings" (lines 11-14, pp.6).	All general practices eligible for our trial are, or will be, drawn from Yorkshire, East Midlands and the North East of England – we do not provide any further information to ensure anonymity of participating practices when the study is written up. We aim to ensure 'a range of multi-ethnic, rural and urban populations to maximise generalisability of findings' in a number of ways. Randomisation is stratified for: practice list size, rural/urban populations (see p6). We are recruiting general practices from ethnically diverse regions, inviting all eligible patients and ensuring that interpretation services are available (Box 2, p6). Finally, our trial steering

			committee will ensure that there is no selection bias related to ethnicity within our (p10).
16		Fourth, further elaboration of how insights from the feasibility study informed the CANAssess especially in collecting the data for the cost-effectiveness analysis is needed unless its contribution to the final design of the protocol/study is minimal.	Thank you, we have added the following text to p6 alongside other examples of how feasibility work influenced design of our trial. “Our Resource Use Questionnaire (RUQ) was also modified following feedback from patient participants in the feasibility study.”
17		A minor point, meanwhile, is to clarify in the protocol whether recruitment has started (https://ctru.leeds.ac.uk/wp-content/uploads/2021/08/CANAssess_Participant_Follow-up-Newsletter_v2.0_20200420_August.pdf)	Thank you, we make it clear in the Trial Status section (p15). We have updated numbers in the section in terms of: “as of 25.01.2021 we have 27 general practices open to recruitment and have recruited 333 patient participants and 102 carer participants.”
18		A last point concerns the conduct of a debriefing with participants including the GPs that will be taking part in the study.	In our dissemination section on p14 we indicate the following. Our change is indicated in highlighted text. “Findings will be presented and discussed at a final dissemination meeting, to which a wide range of stakeholders will be invited, including trial clinicians, participants and those involved in the stakeholder engagement. Results of the study will be published in peer-review publications and will be presented at national and international conferences. A lay summary of our findings will be published on study and organizational websites, sent to participating general practices and will be accessible to participants.”

VERSION 2 – REVIEW

REVIEWER	Merriel, Samuel University of Exeter, College of Medicine & Health
REVIEW RETURNED	24-Feb-2022
GENERAL COMMENTS	Thank you for asking me to review this revised submission. I am satisfied the authors have addressed the reviewers comments, and have nothing further to suggest.